# Contrasting speed and accuracy approaches to measure executive functions in three low- and middle-income countries

**Charlotte Wray**[1], **Alysse J. Kowalski**[2], **Feziwe Mpondo**[3], **Laura Ochaeta**[4], **Delia Belleza**[5], **Ann DiGirolamo**[6], **Rachel Waford**[7], **Linda Richter**[3], **Nanette Lee**[8], **Gaia Scerif**[9], **Alan Stein**[1,10,11], **Aryeh D. Stein**[7]*, **COHORTS**[¶]

1 Department of Psychiatry, the University of Oxford, Oxford, United Kingdom, 2 Nutrition and Health Sciences Program, Laney Graduate School, Emory University, Atlanta, GA, United States of America, 3 DSI-NRF Centre of Excellence in Human Development at the University of the Witwatersrand, Johannesburg, South Africa, 4 Institute of Nutrition of Central America and Panama, Guatemala City, Guatemala, 5 Department of Psychology, School of Arts and Sciences, University of San Carlos, Cebu, Philippines, 6 Georgia Health Policy Center, Georgia State University, Atlanta, GA, United States of America, 7 Hubert Department of Global Health, Rollins School of Public Health, Emory University, Atlanta, GA, United States of America, 8 USC-Office of Population Studies Foundation, Inc., University of San Carlos, Cebu, the Philippines, 9 Department of Experimental Psychology, the University of Oxford, Oxford, United Kingdom, 10 African Health Research Institute, KwaZulu Natal, South Africa, 11 MRC/Wits Rural Public Health and Health Transitions Research Unit (Agincourt), School of Public Health, Faculty of Health Sciences, University of the Witwatersrand, Johannesburg, South Africa

¶ Membership of COHORTS is provided in the Acknowledgments.
* Aryeh.stein@emory.edu

**Data Availability Statement:** The data used in this study are from active cohort studies with previously published recruitment information and

## Abstract

Executive functions (EF) can be measured by tests assessing accuracy, reaction times and by computing scores which combine these two components. Interpretation issues can arise from the use of different scoring methods across studies. Given that EF measures and their scoring methods are predominantly developed and validated in high income countries, little is known about the generalisability of such methods cross- culturally. The current paper compares two different established scoring approaches for measures of inhibition and cognitive flexibility: difference scores (which utilise reaction time only) and computed scores (combining accuracy and reaction time). We utilised data collected in adulthood from three low- and middle-income birth cohorts (Guatemala, Philippines, South Africa). Non-normal distributions were observed for both scoring methods in all three samples; however, this was more pronounced for the difference score method. Differing distribution patterns were observed across the three cohorts, which was especially evident in the Guatemala cohort, highlighting potential issues with using these methods across diverse populations. The data suggest that the computed scores may be a reliable measure of EF. However, the different ways of scoring and interpreting EF instruments need to be considered carefully for each population before use.

hence there is potential for individual participant reidentification if the raw data are made freely available. Data are available to qualified researchers upon request, subject to confidentiality agreements. Data requests can be made to the following: South Africa data: Linda Richter (Linda. Richter@wits.ac.za) or Shane Norris (Shane. Norris@wits.ac.za). Guatemala data: Manuel Ramirez (mramirez@incap.int) or Dina Roche (droche@incap.int). Philippines data: USC Office of Population Studies Foundation (opsfoundation@opsusc.org), Nanette Lee (nanette_rlee@yahoo.com) or Delia Carba (carbadel@yahoo.com).

**Funding:** This study was funded by the Bill and Melinda Gates Foundation (OPP1164115). The funders had no role in study design, data collection and analysis, decision to publish, or preparation of the manuscript.

**Competing interests:** The authors have declared that no competing interests exist.

## Introduction

Inhibition and cognitive flexibility are key components of higher order executive functions (EF) [1–3], routinely measured across a number of key tasks [4–6]. However, the assessment of EF can be challenging, as EFs inherently have two dimensions by which they are measured; speed and accuracy. This raises questions about how best to utilise measures of accuracy and reaction time to fully capture individual differences in EF and highlights potential challenges for interpretation when different scoring methods are used across different studies [7].

Traditionally, EF in older children and adults is measured by reaction time rather than accuracy, due to higher variation in reaction times in comparison to accuracy scores [8]. Such studies compute difference scores, which measure differences in reaction time between different trial types, creating a conflict score for inhibition tasks, and a switch cost score for cognitive flexibility tasks [9, 10]. For example, a conflict score is the difference in reaction time between trials that require inhibition and those that do not. However, these difference scores do not incorporate an assessment of accuracy, which may result in the loss of potentially rich information about individual differences in EF. There have been calls to move away from relying solely on reaction time difference scores, as some suggest that these scores do not produce reliable individual differences [6, 10] and often have poor test-retest reliability [11].

Alternatives to difference scores include scores which combine speed and accuracy data. Proponents of these combined scores suggest that these methods better capture individual differences, in a number of parallel ways, such as creating composite scores [12, 13] or by using latent variable models [12, 13]. However, there is currently no consensus on which scoring method works best, or whether these composite methods do capture EF more precisely than exploring reaction time and accuracy separately. The use of different scoring methods across different studies makes comparisons difficult. The scoring method chosen by researchers could have implications for the interpretability of the findings and the extent to which cross-study comparisons can be made. Reliable measurement of EF is essential in order to fully capture individual differences [7]. Furthermore, the impact of scoring methods may vary depending on population, especially given that EF assessments are usually developed in high income countries. The extent to which different scoring methods consistently capture individual differences in EF cross-culturally needs further investigation.

The current study compared an approach based on difference scores [6, 10] and an approach combining accuracy and speed [14, 15], using data from three diverse low and middle income birth cohorts, which have reached adulthood. An exhaustive treatment of all approaches that treat accuracy and speed separately (e.g., difference scores, 8), as opposed to combining them (e.g., composites, latent factors, 12, 13, is beyond the scope of a single manuscript. However, a comparison could be informative in the context of data that are unique or unusual in representing populations outside the Global North. In particular, we selected as an example of the latter the NIH toolbox computed scores (which combine accuracy and reaction time into a single metric). We utilised the NIH toolbox EF tasks, as these are well established measures of EF, which have been well validated and show good validity and reliability in both adults and children [16–18]. However, as these tasks were developed and validated in the United States, little is known about how the NIH toolbox method for computing a composite speed and accuracy score applies to diverse populations outside of the USA. We aim to highlight the strengths, weaknesses and implications of these different scoring approaches, when used with data from three diverse populations.

## Method

We utilised EF data collected from three large samples, using two of the NIH toolbox cognitive assessments [19], the Flanker Inhibitory Control and Attention task and the Dimensional

Change Card Sort (DCCS), which measures cognitive flexibility. Data were collected in three low- and middle-income (LMIC) birth cohorts in Guatemala, Philippines and South Africa. These cohorts have been prospectively followed throughout their lives and were recently revisited when participants were in adulthood.

## Cohorts

**Institute of Nutrition of Central America and Panama Nutrition Trial Cohort (INCAP), Guatemala.** The INCAP study began in 1969 and originally comprised 2392 children born between from 1962–1977 in eastern Guatemala [20]. Here we report on data collected when cohort participants were 40–57 years old (n = 1271).

**Cebu Longitudinal Health and Nutrition Survey (CLHNS), The Philippines.** The CLHNS recruited a community-based sample of pregnant women in 1983–1984 in Metropolitan Cebu, Philippines. Participants (offspring) were followed up throughout infancy, childhood and adulthood (N = 3080 at recruitment); see Adair et al. [21] for cohort profile. This paper reports data collected when cohort participants were 34–36 years old (n = 1327).

**Birth to twenty plus (Bt20+), Soweto, South Africa.** Established in 1990, the Bt20 + cohort comprised 3273 singleton births enrolled within a 6-week period in public health centres in Soweto and Johannesburg [22]. This paper reports data collected when cohort participants were 28–29 years old (n = 1402).

## Procedure

All participants were assessed by a trained fieldworker, either at a central assessment facility (Guatemala and South Africa) or in their own home (Philippines). Inhibition and cognitive flexibility tasks (NIH Flanker and NIH DCCS, respectively), were administered as part of a larger testing battery that also included measures of working memory (NIH toolbox list sorting), processing speed (NIH toolbox Pattern Comparison) and non-verbal IQ (Raven's Standard Progressive Matrices). All NIH toolbox measures were administered according to the NIH Toolbox manual, using an iPad. The tasks were administered in Spanish in Guatemala, Cebuano in the Philippines and English in South Africa.

Written consent was obtained from all participants before the assessments began. Ethics clearance for the study was obtained from the institutional review boards of Emory University, Atlanta, USA (IRB00095960); INCAP, Guatemala City, Guatemala (CIA-REV 072/2017); University of San Carlos, Cebu, Philippines (006/2018-01); University of the Witwatersrand, Johannesburg, South Africa (M180225); and the University of Oxford, UK (OxTREC: 518–19). Additional information regarding the ethical, cultural, and scientific considerations specific to inclusivity in global research is included in the S1 Checklist.

## Measures

**NIH Toolbox Flanker Inhibitory Control and Attention [23].** The Flanker task measures inhibition and attentional control by asking participants to attend to target stimuli, whilst inhibiting information irrelevant to the task goals. During each trial, participants are presented with a row of five arrows in the centre of the screen and are asked to indicate the direction of the middle arrow, by selecting an icon pointing in the correct direction at the bottom of the screen (Fig 1). Participants are instructed to use their dominant hand and to return their hand to a standardized "home base" between trials. The fieldworker first models the task for the participant. Participants then complete four practice trials to ensure they understand the task and the required response. Participants are required to correctly complete three of the four practice trials to proceed to the test trials. Participants complete 20 test trials in total; twelve congruent

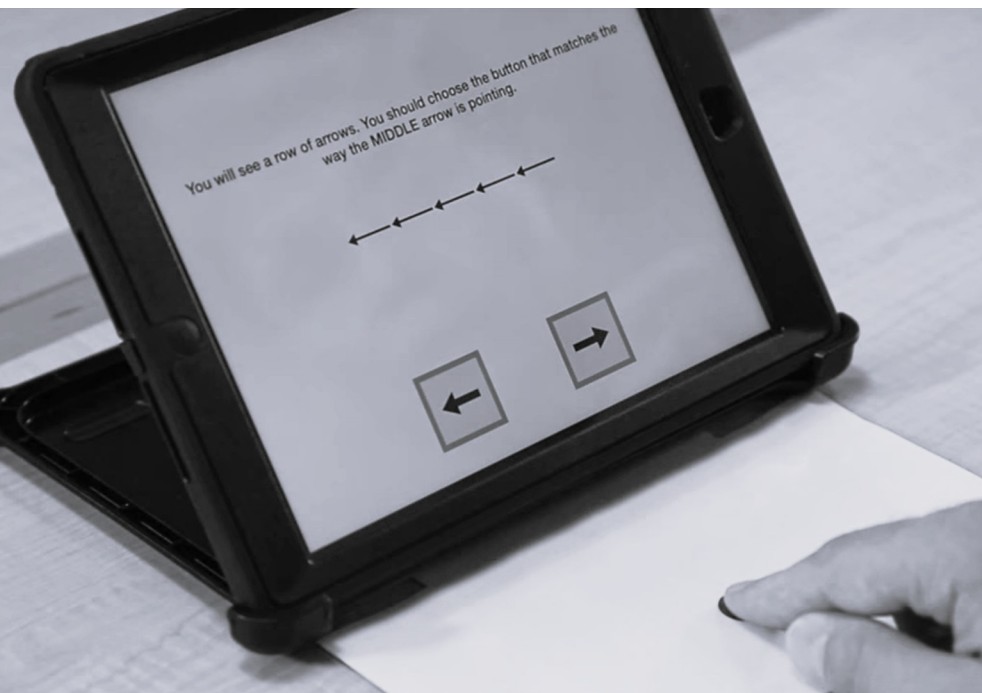

**Fig 1. Screenshot from NIH Toolbox Flanker Inhibitory Control and Attention practice trial.**

trials (the flanking arrows point in the same direction as the middle arrow) and eight incongruent trials (the flanking arrows point in the opposite direction to the middle arrow). The trials occur in the same order for each participant. Accuracy and reaction time are digitally recorded for each trial.

**NIH toolbox Dimensional Change Card sort (DCCS) [23].** The DCCS task measures cognitive flexibility and attention by asking participants to switch between matching pictures by colour and matching pictures by shape. Participants are first presented with two target pictures at the bottom of the screen. Following this, the word 'colour' or 'shape' is presented (orally and visually), to signal to the participant which dimension they should be matching for that trial. The test picture is then presented in the center of the screen, and participants select the target picture which matches either the 'shape' or 'colour' cue of the test picture (Fig 2). Participants use their dominant hand and return to "home base" between trials. The fieldworker first models the task for the participant. Participants then complete four practice trials, first by matching shape and then colour. Participants are required to correctly complete three of the four practice trials to proceed to the test trials. Participants complete 30 test trials, which comprise 23 repeat trials and 7 switch trials. The switch trials occur in the same position for each participant. Accuracy and reaction time were digitally recorded for all trials.

**Reliability.** Reliability data for the NIH Computed Scores and Raven's Standard Progressive Matrices for each of our sites are reported elsewhere as part of a separate test-retest study with samples that resemble our three cohorts [24]. Wray et al. (2020) report intraclass correlations (ICC) which indicate a high degree of reliability for the Flanker Computed Score (ICC: Guatemala = 0.76; Philippines = 0.79; South Africa = 0.68), DCCS Computed Score (ICC: Guatemala = 0.80; Philippines = 0.48; South Africa = 0.64) and Raven's Standard progressive Matrices (ICC: Guatemala = 0.86; Philippines = 0.85; South Africa = 0.69). In addition, for the current paper we have also calculated ICCs for the difference scores. Reliability is more varied across sites for the Flanker Conflict Score (ICC: Guatemala = 0.36; Philippines = 0.69; South

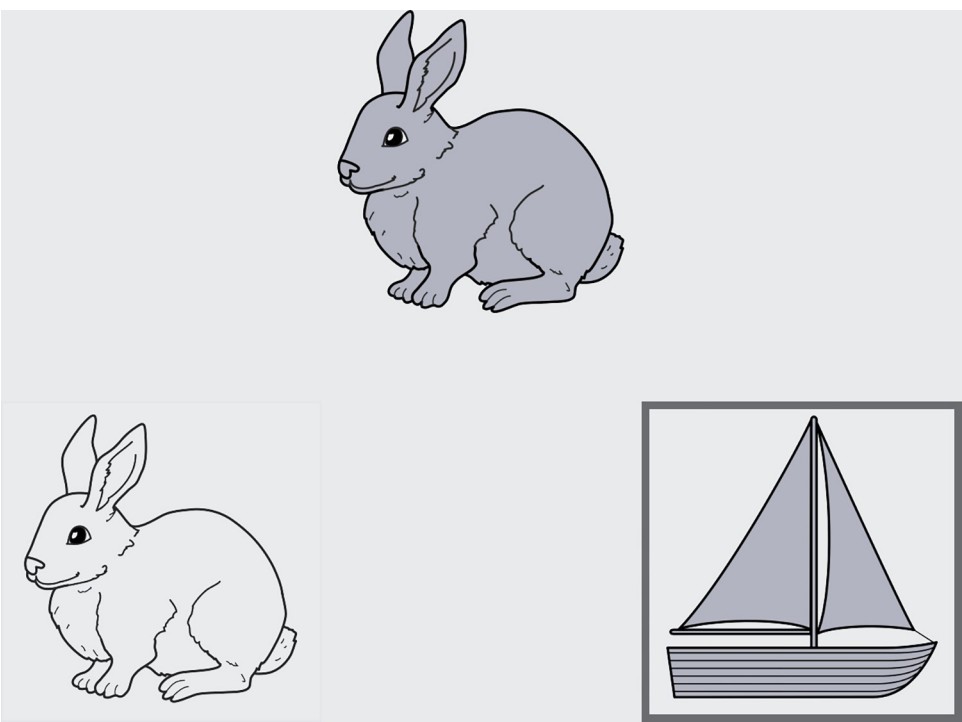

**Fig 2. Screenshot from NIH Toolbox Dimensional Change Card sort practice trial.**

Africa = 0.13) and DCCS Switch Cost Score (ICC: Guatemala = 0.62; Philippines = 0.20; South Africa = 0.39).

**Scoring the NIH toolbox Flanker and DCCS tasks.**   Data from both the Flanker and DCCS tasks comprise reaction time and accuracy scores, for each trial. As such, these tasks can be explored by examining 1) raw accuracy scores, 2) reaction time difference scores, or through 3) computed scores which combine accuracy and reaction time.

**Reaction time difference scores: Flanker conflict scores and DCCS switch cost scores.** Using the raw reaction time data from the NIH toolbox we calculated a *conflict score* for the Flanker task and *switch cost* score for the DCCS task, as described below.

In line with the NIH toolbox scoring manual, reaction time scores less than 100ms or more than 3SD from an individual's mean were excluded when calculating both the Flanker conflict score and DCCS switch cost [25]. Although the NIH toolbox algorithm also truncates reaction time data to a minimum of 0.5 seconds and a maximum to 3 seconds, we did not follow this process for the following calculation, as participants in our cohorts had a wide range in reaction times. This was particularly important for the Guatemala cohort who had median reaction times for incongruent trials ranging from 0.60 seconds to 9.76 seconds. In these instances, truncating our data in this way would have removed meaningfully varied data points from our dataset.

**Flanker conflict score.**   The conflict score measures the difference between the median reaction time for incongruent trials and the median reaction time for congruent trials during the Flanker task and is calculated for correct trials only. The following formula was used:

*Flanker conflict score*
$$= \textit{Median RT for correct incongruent trials} - \textit{Median RT for correct congruent trials}$$

A conflict score cannot be calculated for individuals who did not get any trials correct, or those who scored a zero on either all congruent or all incongruent trials.

**DCCS switch cost.** A switch cost was calculated for the DCCS task, to provide an index of the impact of switching from one requirement (matching by shape) to another requirement (matching by colour).

This was measured as the difference between the median reaction time for correct switch trials and the median reaction time for correct repeat trials. However, due to the different number of repeat trials (n = 23) and switch trials (n = 7), this formula was modified to compute the difference in median reaction time between the switch trials and the repeat trial immediately preceding the switch. The following formula was used for this calculation:

$$\textit{DCCS Switch Cost}$$
$$= \textit{Median RT for correct switch trials}$$
$$- \textit{Median RT for correct repeat preceding the Switch trials}$$

A switch cost score cannot be calculated for individuals who did not get any trials correct, or for those who scored a zero on either all switch or all repeat preceding the switch trials. See Table 1 for difference score means.

**NIH toolbox computed score.** The NIH toolbox uses a two-vector algorithm to score the Flanker and DCCS tasks, which is calculated automatically within the NIH toolbox app and can be exported alongside the raw data. This algorithm utilizes an accuracy vector and a reaction time vector [25] See Table 1 for NIH computed score means.

For this two-vector process, the accuracy vector is calculated first. In addition to the test trials (30 for the DCCS and 20 for the Flanker), participants age 8 and over also receive additional accuracy points for trials that are part of the test for younger children, which research has shown, adults score at ceiling on [25]. This means that in total there are 40 possible accuracy points for the Flanker task (20 test trials plus 20 additional points) and DCCS task (30 test trials and 10 additional points). The accuracy vector is calculated by multiplying the number of correct responses by 0.125 (5 points divided by 40 trials), to give an accuracy score ranging from 0–5.

The reaction time vector is calculated using participants' raw median reaction times. For the Flanker task, this is only calculated for correct incongruent trials and for the DCCS correct switch trials. Reaction time data that are less than 100ms or more than 3SD from an individual's mean reaction time are excluded. A log (base 10) transformation is performed as data distribution for these tasks are usually positively skewed. In line with the NIH toolbox scoring manual [25], this algorithm also sets the minimum median reaction time to 500ms and the maximum to 3000ms. Any values above or below these were truncated to 500ms or 3000ms. Log values were then rescaled to a 0–5 range, such that smaller log reaction time values (faster reaction times) correspond to scores at the upper end of the 0–5 range and larger log values are at the lower end of the range.

**Table 1. Flanker and DCCS: Mean (SD) reaction time difference scores and computed scores.**

|  |  | Guatemala | | Philippines | | South Africa | |
| --- | --- | --- | --- | --- | --- | --- | --- |
|  |  | N | Score | N | Score | N | Score |
| Flanker | Conflict Score | 1155 | 0.50 (1.00) | 1321 | 0.14 (0.37) | 1315 | 0.19 (0.43) |
|  | NIH Computed | 1231 | 5.56 (1.19) | 1327 | 7.48 (1.08) | 1327 | 7.43 (1.11) |
| DCCS | Switch Cost | 1005 | 0.002 (0.56) | 1313 | 0.01 (0.18) | 1312 | 0.06 (0.19) |
|  | NIH Computed | 1223 | 5.26 (1.97) | 1327 | 7.53 (1.29) | 1327 | 7.52 (1.18) |

The computed score is calculated by adding together the accuracy and reaction time vectors, providing a computed score ranging from 0–10. The reaction time vector is only calculated for participants who have an accuracy score of more than 80%. For those participants who do not have an accuracy score of more than 80% their computed score is based solely on their accuracy vector (maximum of 5).

# Results

## Sample characteristics

In Guatemala, mean (SD) age of participants was 47.4 (4.2) y and 44% male. Mean (SD) age of participants in Philippines was 34.5 (0.5) y, 54% were male. In South Africa mean (SD) participant age was 28. 5 (0.4) y, 47% were male.

## Raw accuracy scores

Participants in South Africa and Philippines exhibit high accuracy scores on both tasks, with many participants scoring at ceiling (See Fig 3A and 3B). Many participants from Guatemala also score at ceiling, however more variation in scores is observed in Guatemala for both tasks, which is most pronounced for the DCCS task (Fig 3B).

## Attrition

One key factor in determining the suitability of a scoring mechanism is to explore attrition rates. The Flanker conflict score calculation required participants to correctly respond to at

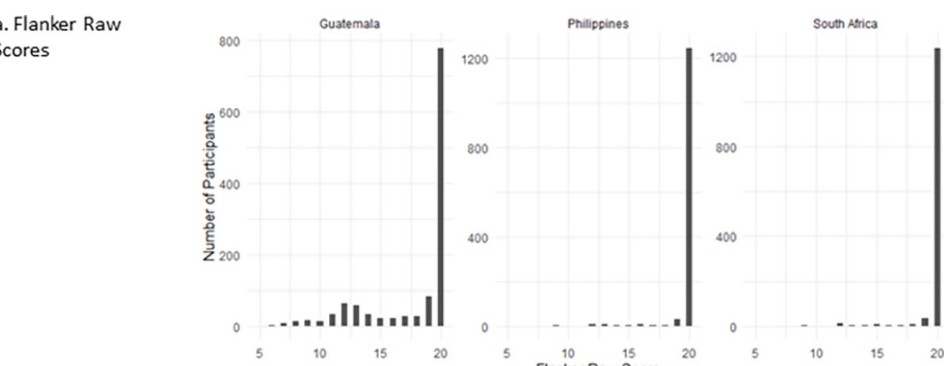

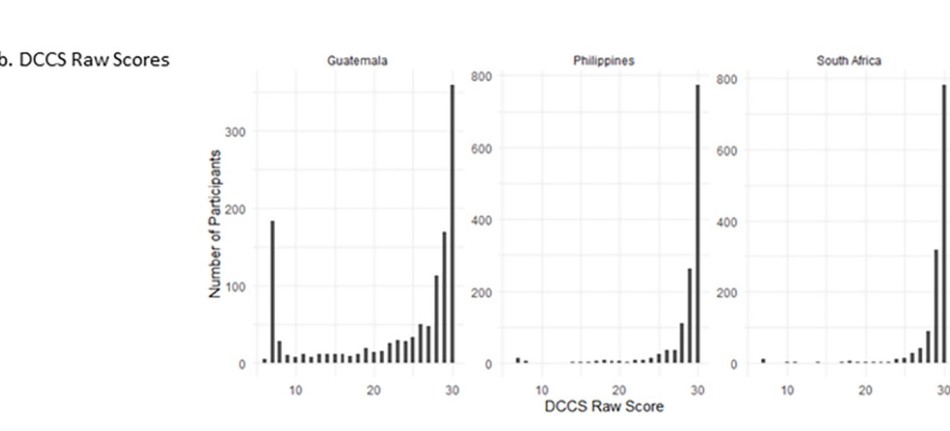

**Fig 3.** Distributions of raw scores for the (a) Flanker and (b) DCCS task in Guatemala, Philippines and South Africa.

least one congruent and one incongruent trial. Similarly, the DCCS switch cost required at least one correct switch trial and one correct repeat trial. As a result, Flanker conflict scores could not be calculated for 58 (4.8%) Guatemalan, 6 (0.5%) Filipino, and 12 (0.9%) South African participants. In addition, DCCS switch cost scores could not be calculated for 190 (15.5%) Guatemalan, 13 (1.0%) Filipino, and 12 (0.9%) South African participants. These data were in essence lost to further calculations (e.g., correlations with other cognitive variables, such as non-verbal IQ).

The NIH toolbox computed score only calculates a reaction time vector if accuracy is above 80%. Within our cohorts, this cut off results in the exclusion of reaction time data from a large number of participants. For the Flanker task, 292 Guatemalan participants, 40 Filipino participants and 37 South African participants did not meet the cut off requirement. The cut off had a greater impact on the DCCS task, during which 452 Guatemalan participants, 84 Filipino participants and 54 South African participants did not meet the 80% accuracy cut off. However, as the computed scores in cases of accuracy below 80% are calculated on the basis of accuracy alone, those data were not lost to further calculations.

## Distributions

Next, we considered the distributions for each of the scoring methods across sites (see Figs 4 and 5). Table 2 indicates that the data are not normally distributed for either computed or difference scores. The DCCS switch cost score distributions are highly skewed in all three sites (see Fig 4A). The NIH DCCS computed score also reveals highly skewed distributions for the

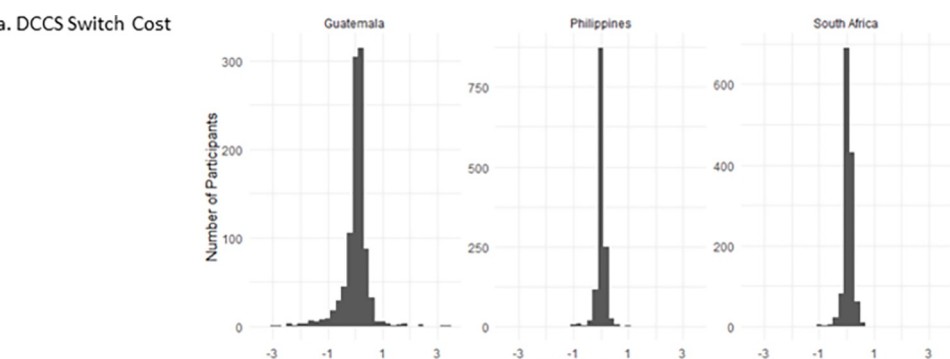

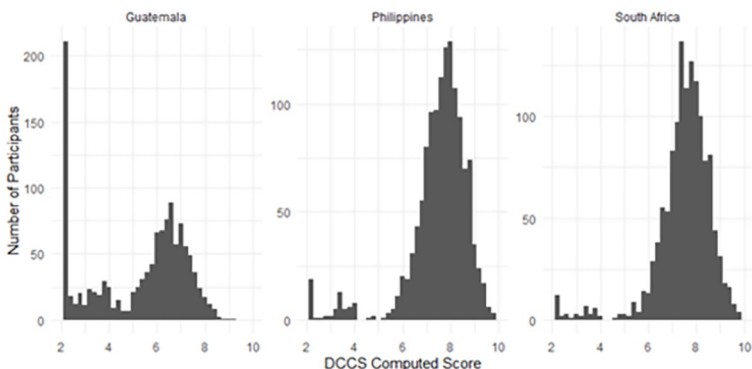

**Fig 4.** Distributions of (a) DCCS Switch Cost Scores and (b) DCCS NIH Computed Scores in Guatemala, Philippines and South Africa. Note: DCCS switch cost represents the difference in median response time between correct switch trials and correct repeat preceding the switch trials, in seconds.

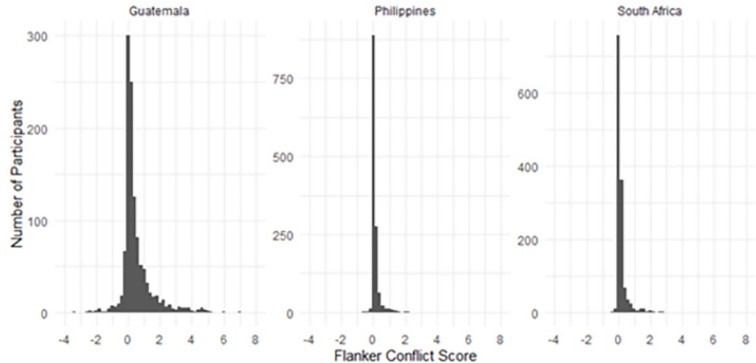

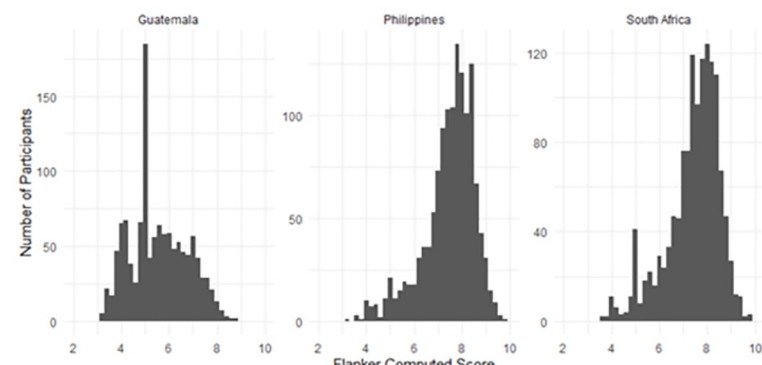

**Fig 5.** Distributions of (a) Flanker Conflict Scores and (b) Flanker NIH Computed Scores in Guatemala, Philippines and South Africa. Note: Flanker conflict scores represent the difference in median response time between correct incongruent trials and correct congruent trials, in seconds.

Philippines and South Africa; however, the Guatemala distribution is only minimally skewed (see Fig 4B).

Flanker conflict score distributions are highly skewed in all three countries (see Fig 5A). When using the NIH Flanker computed score however, the Philippines and South Africa are moderately skewed, with the Guatemala data indicating minimal skew (see Fig 5B). Overall, the NIH computed distributions are less skewed than are the difference scores.

In addition, Fig 3B highlights a 'spike' in DCCS raw scores at '7' for the Guatemala sample, which is not observed in the distribution of the other cohorts. This 'spike' in the Guatemala raw accuracy scores is reflected in the NIH computed score, as demonstrated by the high number of participants with a computed score of 2.13 (See Fig 2B). Of the 183 (15%) participants who scored seven, all but one participant scored zero on the shape (repeat) trials but answered all seven colour (switch) trials correctly. Further investigation of these individuals indicated that on average they completed more practice trials than those that did not score 7. In addition,

**Table 2. Skewness and Kurtosis for each scoring method across countries.**

|  | Guatemala | | Cebu | | South Africa | |
|---|---|---|---|---|---|---|
|  | Skew | Kurtosis | Skew | Kurtosis | Skew | Kurtosis |
| Flanker Computed | 0.26 | 2.25 | -1.06 | 4.24 | -0.95 | 3.73 |
| Flanker Conflict Score | 2.29 | 12.17 | 8.07 | 101.1 | 6.49 | 69.85 |
| DCCS Computed | -0.46 | 1.80 | -1.94 | 8.27 | 1.77 | 8.65 |
| DCCS Switch Cost | -2.57 | 29.27 | -2.53 | 25.92 | -2.74 | 35.34 |

**Table 3. Associations between each of the measures in Guatemala.**

| Guatemala | RSPM | Flanker Conflict Score | DCCS Switch Cost | Flanker Computed Score |
|---|---|---|---|---|
| Flanker Conflict Score | -.15** | - | | |
| DCCS Switch Cost Score | .09** | -.07** | - | |
| Flanker Computed Score | .45** | -.37** | .10** | - |
| DCCS Computed Score | .50** | -.18** | .21** | .54** |

Note

*p < .05

**p<0.01, RSPM: Raven's Standard Progressive Matrices

those who had a raw score of 7 were slightly older (48y vs 47y, *p*<0.001, *d* = 0.27), generally completed fewer grades of schooling (3y vs. 5y, *p*<0.001, *d* = 0.87), and had lower non-verbal IQ, as measured by the Raven's Standard Progressive Matrices (Mean: 13.14 vs. 17.06, *p*<0.001, *d* = 0.81), than other participants.

A peak in Flanker computed scores is also observed in the Guatemala data at 'five points', which again is not seen in the other cohorts (See Fig 5B). Further exploration of participants who scored 'five' computed score points on the Flanker task indicate that their accuracy scores were at ceiling (Mean: 19.99, SD: 0.08); however, their reaction times were relatively slower than other participants. Participants who scored five computed points had response times of 2.53 seconds on average, which was significantly slower than those that did not score five points (Mean: 1.82 seconds, *p* <0.001, *d* = 0.86).

### Correlation between scoring approaches and non-verbal IQ

Finally, we explored the relationships between the NIH computed scores, the difference scores and the Ravens Standard Progressive Matrices, to better understand the associations between each scoring method and non-verbal IQ.

Across all cohorts, the Flanker computed score is significantly associated with the Flanker conflict score. In addition, the DCCS computed score is significantly correlated with the DCCS switch cost score, positively in Guatemala and Philippines, and negatively in South Africa. The difference in direction of these correlations was surprising, but perhaps dependent on the computed score being more influenced by accuracy for the first two samples, and by reaction time differences (in the context of overall high accuracy) for the South African sample. However, we note that these associations are of small magnitude (See Tables 3–5).

Across all cohorts non-verbal IQ is significantly associated with the Flanker conflict score, Flanker computed score and DCCS computed score. In both Guatemala and Philippines, non-verbal IQ is also weakly associated with the DCCS switch cost score. However, no such

**Table 4. Associations between each of the measures in the Philippines.**

| Philippines | RSPM | Flanker Conflict Score | DCCS Switch Cost | Flanker Computed Score |
|---|---|---|---|---|
| Flanker Conflict Score | -.23** | - | | |
| DCCS Switch Cost Score | .06* | -.12** | - | |
| Flanker Computed Score | .41** | -.54** | .10* | - |
| DCCS Computed Score | .45** | -.26** | .08* | .57** |

Note

*p < .05

**p<0.01, RSPM: Raven's Standard Progressive Matrices

**Table 5. Associations between each of the measures in South Africa.**

| South Africa | RSPM | Flanker Conflict Score | DCCS Switch Cost | Flanker Computed Score |
|---|---|---|---|---|
| Flanker Conflict Score | -.28** | - | | |
| DCCS Switch Cost Score | -.03 | -.04 | - | |
| Flanker Computed Score | .42** | -.63** | .00 | - |
| DCCS Computed Score | .39** | -.38** | -.06* | .54** |

Note

*p < .05

**p<0.01, RSPM: Raven's Standard Progressive Matrices

significant associations were observed in South Africa. The associations between non-verbal IQ and the difference scores are very weak across all countries. The NIH computed scores on the other hand, have much stronger associations with non-verbal IQ (see Tables 3–5).

## Discussion

In this paper we utilised data from three diverse, low- and middle-income populations to explore different approaches to scoring two commonly used measures of EF. Overall, our data indicate that on assessments of inhibition and cognitive flexibility, the NIH computed scores result in less participant attrition than difference scores, enabling more participant data to be retained for further analyses. Non-normal distributions were observed for both scoring methods; however, this was more pronounced for the difference scores. The NIH computed scores and difference scores were associated with non-verbal IQ across all three sites, and these associations were strongest for the NIH computed scores.

### Accuracy

Participants in South Africa and Philippines exhibit high accuracy scores for both tasks, with many participants scoring at ceiling. This is in line with previous studies suggesting that reaction time data is often a more reliable indicator of individual differences in executive function in young- and mid- adulthood in HIC settings [8]. In the Guatemala cohort, many participants also exhibited high accuracy scores. However, more variation was observed.

### Attrition

One clear advantage of the NIH computed scores is that they enable the retention of all participants. The difference scores on the other hand result in a loss of participant data, as they can only be calculated if a participant has at least one switch and repeat trial correct during the DCCS task (congruent and incongruent trial for the Flanker task). For populations with high accuracy rates, attrition is minimal. However, in the Guatemala cohort, where accuracy rates were the most variable, greater data loss was observed. This highlights a potential limitation of difference score methods when used across populations with varying ages and educational experiences, which may influence task performance. Furthermore, as our cohorts show differing variation in accuracy scores, a measure of EF that uses solely reaction time, may not capture all individual differences associated with executive function.

### Distribution

Distributions for both computed scores and difference scores were skewed across the three cohorts. Overall, the NIH computed score distributions were less skewed than for the

difference scores. However, of note is that a log (base 10) transformation is used in the computed score calculation and thus some skewness has already been addressed.

In addition, a 'spike' in DCCS raw scores at '7' was observed for the Guatemala sample, which was also reflected in the NIH computed score. Nearly all participants who scored seven, scored zero on the shape (repeat) trials but answered all seven colour (switch) trials correctly. As outlined above, during the DCCS task, participants are first asked to match by 'shape' (repeat trials) and then at seven points are asked to match by 'colour' (the switch trials). This indicates that these participants matched by colour throughout the whole task, even on trials which cued them to match by shape. This may be an indication that this group of participants did not fully understand the instructions for the task, despite having passed the practice test trials. On average this group did complete more practice trials than those that did not score 7 (the practice trials were designed so that more practice trials were given if a participant got practice trials incorrect). Further investigation of these individuals indicated that those who had a raw score of 7 were slightly older, generally completed fewer grades of schooling, and had lower non-verbal IQ, than other participants.

In addition, a peak in Flanker NIH computed scores was observed in the Guatemala data at '5 points', which again is not seen in the other cohorts. Participants who scored 'five' computed score points on the Flanker generally had raw accuracy scores which were at ceiling, but relatively slow reaction time scores. This suggests that these participants responded slowly, but accurately and as a result, their computed score comprises a high accuracy vector, but low reaction time vector. This raises the question about the impact that task instructions have on participant response behaviours. For example, in cultures or age groups that prioritise accuracy over speed, the instruction "respond as quickly as you can" may be followed to a different extent. It is difficult to determine the extent to which participants follow task instructions and whether participants prioritise accuracy over speed or vice versa. However, this is a particularly important point to highlight given the diversity in culture and ages across our cohorts and may explain some of the differences observed across cohorts. This limitation would equally impact both reaction time scores and combined scores.

The unusual distributions observed in Guatemala were only observed when using the NIH computed scores and not the difference scores. This highlights that the scoring methods used for these tasks may have implications for the interpretation of findings, especially in LMIC contexts, where more varying cultural, socio-economic, and educational experiences may result in more varied performance.

## Associations between scoring methods and non-verbal intelligence

Across all cohorts the Flanker computed score was weakly associated with the Flanker conflict score and the DCCS computed score was weakly associated with the DCCS switch cost score. Given the different methodologies of these scores, these weak associations are not wholly unexpected. In addition, some unexpected correlations also emerged, again with difference scores; the DCCS conflict score was positively associated with RSPM, whereas the Flanker Conflict score correlated negatively with it. This may suggest a divergence between the two (reaction time based) difference scores and how they relate to individual differences in non-verbal intelligence. Although we can only speculate on interpretations, it is possible that the differences in relations to RSPM depend on different levels of overall cognitive load and average speed imposed by the two tasks. Ultimately, it is very hard to disentangle alternative explanations on the basis of difference scores alone, because they lack a true baseline for overall difficulty or cognitive load.

As a whole, while both scoring methods were related to non-verbal intelligence, the strength of associations was much stronger with the NIH computed scores. This suggests that the NIH

computed score may be a more reliable measure of executive functioning than the difference scores, which indicated only a weak association with non-verbal intelligence.

Considerations for the use of combined speed/ accuracy indices: The case of NIH Toolbox computed scores.

A key broader point emerging from the comparison of two scoring methods, is that, if both accuracy and reaction time convey unique information to EF ability, they could be combined. The benefit of combined scores applies to the broader class of combined scores like the two vector-scores solution by NIH Toolbox, but also other approaches, such as latent variable models [12, 13]. While a comparison of all combined methods was beyond the scope of a single manuscript, it is really worth noting that, as a class of methods, they outperform difference scores along many dimensions.

In addition, and perhaps most helpful in a population health setting, the NIH Toolbox approach has a practical advantage over latent variable models:–it just applies an algorithm rather simply, which in applied and practical terms is more helpful than collecting data on the whole sample, and then investigating the latent variable structure of the data. A further benefit is that individual differences across multiple underlying cognitive processes feeding into both accuracy and speed are well reflected by combined scores. This is important for researchers interested in capturing cognitive underpinnings of individual differences. However, one advantage of a latent score over the NIH Toolbox approach is that latent scores empirically determine how strongly to weight each item, whereas each item is weighted equally in the NIH Toolbox score. As such the latent variable score may be a more precise measurement of individual differences.

Although we have highlighted strengths of the NIH toolbox computed scores, and these strengths are in line with those suggested for computed scores more generally [6], there are some limitations, which need to be considered before future studies utilise them for measuring EF.

One consideration is the 80% cut off used when calculating the NIH computed score; this calculation only uses reaction time data if accuracy scores are greater than 80%. As a result, for those participants with accuracy of less than 80%, the Flanker and DCCS scores are solely based on accuracy and not a combination of accuracy and reaction time. For most adult populations, this cut off would not be an issue, as adults generally score highly on this task. Indeed only 3–6% of participants did not meet this cut off in the Philippines and South Africa cohorts. However, in Guatemala, 24% of participants did not meet this cut off for the Flanker task, and 37% for the DCCS task. Thus, in populations where a greater number of people do not meet this cut-off, the extent to which the NIH computed score provides an accurate measure of individual differences in executive function may be limited. Alternative scoring methods, which measure and combine accuracy and reaction time individually, may be more appropriate [see 6, 10], to ensure that variation in EF is fully captured.

One further consideration is that the NIH computed score does not differentiate between trial type. When calculating the computed score, the accuracy vector takes all trials into account. The reaction time vector is calculated using participants' raw median reaction times for correct <u>incongruent</u> trials (Flanker) and for correct <u>switch</u> trials (DCCS). However, in this instance we cannot ascertain whether a low reaction time score is a result of participants' difficulties with inhibition (or task switching), or simply reflects slow reaction time generally. As a result, it is difficult to understand the mechanism behind the variation in scores when using the NIH computed score. Although the *difference scores* (DCCS switch cost and Flanker conflict scores) indicate less variation than the NIH toolbox computed scores, they reveal more information about the unique impact of inhibition and task switching on performance. For example, the Flanker conflict score indicates the impact of inhibiting irrelevant information

(incongruent trials) on task performance, in comparison to trials when no inhibition is required (congruent trials). By distinguishing between the two types of trials, *difference scores* allow us to determine the direct cost of inhibition/task switching, rather than simply measuring processing speed and/or attention.

One final consideration is that the NIH toolbox tasks have very few trials (20 in the Flanker and 30 in the DCCS task), which is unusual for these type of tasks, which typically have around 50 per condition. The number of trials is a limiting factor as they may impact the extent to which they can reliably measure both accuracy and reaction time. However, this is a limitation of the design of the task, rather than scoring method, as this issue would impact both computed score and difference scores for the NIH toolbox tasks.

Although the NIH toolbox computed scoring method has many advantages, the potential different ways of scoring and interpreting the NIH Toolbox instruments need to be carefully thought through before using the NIH toolbox, especially with diverse populations.

Overall, our data suggest that the NIH toolbox computed scores lead to the least exclusion of participant data, are more normally distributed, and have stronger correlations with another well- validated measure of cognition, than the more traditional *difference score* methods. Our observation supports other papers which highlight the advantage of such computed methods, combining accuracy and reaction time, in capturing variation in EF [7]. However, differing distributions were observed across our three cohorts, which were especially evident in the Guatemala cohort, highlighting potential issues with using these methods across diverse populations. Through assessing how each scoring method operates in three diverse samples, we highlight strengths, weaknesses and implications that may not have been as apparent had we only measured data from adults in more homogeneous and affluent contexts. Each of our LMIC cohorts has different characteristics, languages, ages and backgrounds, which may explain why differing patterns across countries were observed. As a result, the scoring and interpretation of these measures needs to be considered carefully across diverse settings.

## Supporting information

**S1 Checklist. Inclusivity in global research.**
(DOCX)

## Acknowledgments

We would like to thank all of the participants in this study. With thanks to all members of the Consortium of Health-Orientated Research in Transitioning Societies (COHORTS) group. Additional members of the COHORTS group include: **Pelotas Birth Cohorts**: Fernando C Barros, Federal University of Pelotas; Fernando P Hartwig, Federal University of Pelotas; Bernardo L Horta, Federal University of Pelotas; Ana M B Menezes, Federal University of Pelotas; Joseph Murray, Federal University of Pelotas; Fernando C Wehrmeister, Federal University of Pelotas; Cesar G Victora, Federal University of Pelotas; **Birth to Twenty Plus**: Shane A Norris, University of the Witwatersrand; Lukhanyo Nyati, University of the Witwatersrand; **New Delhi Birth Cohort**: Santosh K Bhargava, Safdarjang Hospital and Vardhman Mahavir Medical College; Caroline HD Fall, University of Southampton; Clive Osmond, University of Southampton; Harshpal Singh Sachdev, Sitaram Bhartia Institute of Science and Research; **INCAP Nutrition Supplementation Trial Longitudinal Study:** Maria F. Kroker-Lobos, INCAP Institute of Nutrition of Central America and Panama; Reynaldo Martorell, Emory University; Manuel Ramirez-Zea, INCAP Institute of Nutrition of Central America and Panama; **Cebu Longitudinal Health and Nutrition Survey:** Linda S. Adair, University of North Carolina at

Chapel Hill; Isabelita Bas, Office of Population Studies Foundation, University of San Carlos; Delia Carba, Office of Population Studies Foundation, University of San Carlos; Tita Lorna Perez, Office of Population Studies Foundation, University of San Carlos. The contact person for this consortium paper is Aryeh D. Stein (aryeh.stein@emory.edu).

## Author Contributions

**Conceptualization:** Charlotte Wray, Alysse J. Kowalski.

**Formal analysis:** Charlotte Wray, Alysse J. Kowalski.

**Funding acquisition:** Linda Richter, Nanette Lee, Alan Stein, Aryeh D. Stein.

**Methodology:** Charlotte Wray, Alysse J. Kowalski.

**Project administration:** Charlotte Wray, Feziwe Mpondo, Laura Ochaeta.

**Supervision:** Linda Richter, Gaia Scerif, Alan Stein, Aryeh D. Stein.

**Visualization:** Charlotte Wray, Alysse J. Kowalski.

**Writing – original draft:** Charlotte Wray.

**Writing – review & editing:** Charlotte Wray, Alysse J. Kowalski, Feziwe Mpondo, Laura Ochaeta, Delia Belleza, Ann DiGirolamo, Rachel Waford, Linda Richter, Nanette Lee, Gaia Scerif, Alan Stein, Aryeh D. Stein.

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
