## [Decision Letter · Decision Letter 0]

18 Nov 2022

PONE-D-22-02319Contrasting speed and accuracy approaches to measure executive functions in three low-and middle-income countriesPLOS ONE

Dear Dr. Wray,

Thank you for submitting your manuscript to PLOS ONE. After careful consideration, we feel that it has merit but does not fully meet PLOS ONE’s publication criteria as it currently stands. Therefore, we invite you to submit a revised version of the manuscript that addresses the points raised during the review process.

You will see from the reviewers' comments that there are some concerns relating to methodology (mainly) as well as some clarity / depth of information on more conceptual issues. Please consider the comments carefully and make sure that you also revise the manuscript, following any changes to data / statistical analysis, to ensure any changes in the findings are appropriately reflected and discussed in the manuscript throughout.  

We look forward to receiving your revised manuscript.

Kind regards,

Theodoros M. Bampouras

Academic Editor

PLOS ONE

“We would like to thank all of the participants in this study. This study was funded by the Bill and Melinda Gates Foundation (OPP1164115).”

“This study was funded by the Bill and Melinda Gates Foundation (OPP1164115). The funders had no role in study design, data collection and analysis, decision to publish, or preparation of the manuscript.”

4. One of the noted authors is a group or consortium [COHORTS]. In addition to naming the author group, please list the individual authors and affiliations within this group in the acknowledgments section of your manuscript. Please also indicate clearly a lead author for this group along with a contact email address

Reviewers' comments:

Reviewer's Responses to Questions

**Comments to the Author**

1. Is the manuscript technically sound, and do the data support the conclusions?

Reviewer #1: Partly

Reviewer #2: Partly

2. Has the statistical analysis been performed appropriately and rigorously? 

Reviewer #1: No

Reviewer #2: Yes

3. Have the authors made all data underlying the findings in their manuscript fully available?

Reviewer #1: No

Reviewer #2: Yes

4. Is the manuscript presented in an intelligible fashion and written in standard English?

Reviewer #1: Yes

Reviewer #2: Yes

5. Review Comments to the Author

Reviewer #1: Thank you for the opportunity to review this very interesting manuscript. There has been increased attention to the use of EF tasks in low- and middle-income countries as well as to different methods of EF task scoring that jointly utilize accuracy and speed. In this manuscript, the authors compare two different scoring methods: RT difference scores and NIH Toolbox two-vector scores.

1. Why did the authors decide to only compare these two types of scoring methods? Recently, latent variable models that make joint use of accuracy and RT data have been proposed (e.g., Camerota, Willoughby, Magnus, & Blair, 2020; Magnus, Willoughby, Blair, & Kuhn, 2017). These seem like they could be particularly useful given the range of accuracy across the three countries studied.

2. Have the NIH Toolbox EF tasks been validated for use in low- and middle-income countries? This seems important to note for this study, especially since some of the participants in Guatemala seemed to misunderstand the task directions.

3. It seems like some of the weaknesses of the difference scores could be remedied by log transforming reaction time prior to calculating the scores. The authors note that the NIH Toolbox does perform a log transformation prior to calculating scores. Doing so for the difference scores would help make a fairer comparison of the two methods, especially given the deviations from normality observed for the difference scores.

4. It might help the reader to call the difference scores “Flanker Conflict Score” and “DCCS Switch Score” as I had a hard time remembering which task the conflict and switch scores were from.

5. In the attrition section (page 12), the authors list the number of participants for whom scores could not be calculated. It would be helpful to show these as percentages of the full samples. Similarly, for the statement on page 13 that “nearly all participants who scores seven…” it would be helpful to know how many scored 7 overall (raw number and percentage of the cohort) and of these, how many scored zero on the shape trials (raw number and percentage).

6. The DCCS switch score was positively correlated with the DCCS computed score in Guatemala and Philippines but negatively correlated in the South African cohort. Why would this be? Also why is the Flanker conflict score negatively correlated with RSPM scores while the DCCS conflict score is positively correlated? This seems to suggest that these two difference scores are telling us different things.

7. The summary of results on page 15 does not match the tables. Non-verbal IQ was associated with switch scores in Guatemala and the Philippines, but not in South Africa.

Reviewer #2: Tests of executive functions (EF) are most frequently evaluated based on different scores describing reaction time and/or accuracy of performances. Results and interpretations depend on the chosen scoring method. The presented study evaluates two scoring methods of inhibition and cognitive flexibility assessed via the NIH Toolbox across three large-scale samples from different countries. Results highlight differences in these methods and possible implications are discussed.

The right choice of measurement is an important issue researchers face when conducting studies and evaluating data. Therefore, comparing strengths and weaknesses of possible scoring methods can prove valuable for future studies. However, I find the focus of the presented comparison to miss critical aspects enabling that choice and have a few additional concerns.

Major Points:

1. I find the introduction and discussion of the scores regarding their theoretical and specific cognitive implications to be a little scarce. Given that the study’s aim is to compare the suitability of scores across populations the theoretical conclusions enabled based on both kinds of scores should also be considered. This is touched upon in one paragraph of the discussion but could be further expanded and introduced. Considerations about conclusions regarding cognitive functionality, etc. provide valuable information to researchers since the study of interindividual differences might not always be the focus for applications.

2. I do not fully understand the reasoning behind the correlation between scores and non-verbal IQ. EFs are linked to general intelligence. However, I would not use an associated construct, i.e., intelligence, to examine whether scores validly reflect a different construct, i.e., EF, as the authors seem to conclude.

3. The results of the Guatemala cohort, as the authors hypothesize themselves, are likely due to participants performing the task wrong. For one sixth of participants the switch cost score could not be computed. Solely based on the computation criteria of the scores the DCCS Computed Score can still be calculated. However, I am unsure whether this should be the case since results reflect performance in a different task, i.e., there is no switching component. Results in and interpretations about this cohort are thus substantially influenced by a group of participants who did not perform the task. Maybe as control, results without this subgroup should be considered.

Minor Points:

1. A suggestion related to my first major point: A key aspect of the study is the benefit provided by scores including both accuracy and speed information rather than just one of the above. A lot of the differences found between scores in this study can be boiled down to this distinction. Although the chosen scoring methods are examples of both cases, I feel that the discussion could benefit from a more general discussion of combined speed and accuracy measures.

2. Although the sample it described elsewhere a little more information like mean/ median age and SD or gender distributions would be informative.

3. The descriptions of the task procedures are rather unspecific. For instance, in the Flanker task, where are the icons presented and are they mapped to different hands? In the DCCS, is stimulus presentation sequentially?

4. Table 1 is missing information on which information is presented in parentheses (SD/ SEM/ IQR)?

5. The bins of the histograms of Figures 2 and 3 for the difference scores differ between cohorts and seem coarse given that they present reaction time data which often depict and find much smaller differences on the level of milliseconds.

6. The labels of the graphs describing the difference scores could reflect more clearly what is depicted, i.e., difference between respective conditions in seconds.

7. Could you explain in more detail how the reaction times are rescaled in the computed scores/ state more clearly which score from 0-5 represents which direction of reaction times? I was a little confused by the inversion appearing in the discussion (0 = slow, 5 = fast)

6. PLOS authors have the option to publish the peer review history of their article (what does this mean?). If published, this will include your full peer review and any attached files.

Reviewer #1: No

Reviewer #2: **Yes: **Lukas Recker

---

## [Author Response · Author response to Decision Letter 0]

8 May 2023

Comments to the Author

Reviewer #1: Thank you for the opportunity to review this very interesting manuscript. There has been increased attention to the use of EF tasks in low- and middle-income countries as well as to different methods of EF task scoring that jointly utilize accuracy and speed. In this manuscript, the authors compare two different scoring methods: RT difference scores and NIH Toolbox two-vector scores.

1. Why did the authors decide to only compare these two types of scoring methods? Recently, latent variable models that make joint use of accuracy and RT data have been proposed (e.g., Camerota, Willoughby, Magnus, & Blair, 2020; Magnus, Willoughby, Blair, & Kuhn, 2017). These seem like they could be particularly useful given the range of accuracy across the three countries studied.

Response: We thank the Reviewer for suggesting reference to latent variable models making joint use of accuracy and reaction time data. In part because of Covid-related delays, the 2020 reference had not been published at the time of submission. We can now include the references to both pieces of work in the main manuscript (p. 3-4 in the introduction and p. 20 in the discussion), as very interesting examples of allied approaches to those that, like the NIH Toolbox two-vector scores, do not treat accuracy and reaction time in isolation. We have now explicitly stated that there are multiple classes of related approaches combining accuracy and RT (p. 3-4), to which we refer the reader, although it was beyond the scope of our manuscript to contrast all of the existing approaches that combine accuracy and RT.

The broader points, that both accuracy and reaction time convey unique information to EF ability, but that it can be combined, applies to both latent variable models and combined scores like the two vector-scores. While we did not use a latent variable approach, the two-vector scoring methodology does take both accuracy and RT into account, so in this sense the two approaches are related. However, this latent variable approach was not considered at the time, as it was published after the manuscript was submitted and because our goal was not to provide a fully exhaustive comparison of all possible approaches to these data. 

In addition, and perhaps helpful in a population health setting, the NIH Toolbox approach has a practical advantage over latent variable models: it applies an algorithm rather simply, which in applied and practical terms is more helpful than collecting data on the whole sample, and then investigating the latent variable structure of the data. We have now referred readers to this advantage in the manuscript (p. 20), as some readers may be particularly interested in applied considerations in the context of public health research. Similarly we see that the scoring is related to criterion measures in ways comparable to traditional scoring approaches. 

We now state: 

‘A key broader point emerging from the comparison of two scoring methods, is that, if both accuracy and reaction time convey unique information to EF ability, they could be combined. The benefit of combined scores applies to the broader class of combined scores like the two vector-scores solution by NIH Toolbox, but also other approaches, such as latent variable models (12,13). While a comparison of all combined methods was beyond the scope of a single manuscript, it is really worth noting that, as a class of methods, they outperform difference scores along many dimensions. 

In addition, and perhaps most helpful in a population health setting, the NIH Toolbox approach has a practical advantage over latent variable models: – it just applies an algorithm rather simply, which in applied and practical terms is more helpful than collecting data on the whole sample, and then investigating the latent variable structure of the data. A further benefit is that individual differences across multiple underlying cognitive processes feeding into both accuracy and speed are well reflected by combined scores. This is important for researchers interested in capturing cognitive underpinnings of individual differences.’

2. Have the NIH Toolbox EF tasks been validated for use in low- and middle-income countries? This seems important to note for this study, especially since some of the participants in Guatemala seemed to misunderstand the task directions.

Response: The NIH Toolbox was developed and validated from a diverse set of over 11,000 participants in the United States and has good validity and reliability properties across a wide age range from child- to adulthood. The NIH Toolbox EF tasks have not been validated for use in low- and middle-income countries. This was beyond the scope of our current project, however we set out to explore the implications of different scoring approaches, including the “off the shelf” composite measure that is calculated in the NIH Toolbox application, in three different populations. However, we do have another paper examining the test-retest reliability and the structure of EF more generally and its convergent validity with factors generally accepted to be associated with EF within our LMIC cohorts (see Wray et al. 2020).

3. It seems like some of the weaknesses of the difference scores could be remedied by log transforming reaction time prior to calculating the scores. The authors note that the NIH Toolbox does perform a log transformation prior to calculating scores. Doing so for the difference scores would help make a fairer comparison of the two methods, especially given the deviations from normality observed for the difference scores.

Response: This suggestion would improve the normality of the difference score distribution but would not address other concerns related to attrition and associations with non-verbal intelligence. It would also make the interpretation more challenging. Log difference scores could be used to rank order individuals, but the values themselves would be uninterpretable. 

4. It might help the reader to call the difference scores “Flanker Conflict Score” and “DCCS Switch Score” as I had a hard time remembering which task the conflict and switch scores were from.

Response: Thank you for this suggestion, we made the change throughout. 

5. In the attrition section (page 12), the authors list the number of participants for whom scores could not be calculated. It would be helpful to show these as percentages of the full samples. Similarly, for the statement on page 13 that “nearly all participants who scores seven…” it would be helpful to know how many scored 7 overall (raw number and percentage of the cohort) and of these, how many scored zero on the shape trials (raw number and percentage).

Response: We added these details to the attrition section and to the distribution section regarding the participants who scored 7. 

6. The DCCS switch score was positively correlated with the DCCS computed score in Guatemala and Philippines but negatively correlated in the South African cohort. Why would this be? Also why is the Flanker conflict score negatively correlated with RSPM scores while the DCCS conflict score is positively correlated? This seems to suggest that these two difference scores are telling us different things.

Response: The negative association in South Africa did not reach significance. Therefore, we hesitate to dwell on the distinction between the correlations across countries, especially given how well statistically powered to detect even very small effects our samples were. However, the second observation, that the DCCS conflict score was positively associated with RSPM, whereas the Flanker Conflict score correlated negatively with it, meets statistical criterion and is intriguing. We agree with the reviewer that this suggests a divergence between the two (reaction time based) difference scores and how they relate to individual differences in non-verbal intelligence. Although we can only speculate on this, it is possible that the differences in relations to RSPM depend on different levels of overall cognitive load / average speed for the two tasks. In other words, DCCS may be more cognitively taxing as a whole across participants compared to Flanker, with individually larger DCCS switch scores index driven by faster reaction times in repeated trials (compared to switch trials). In turn, these higher facilitation effects may associate (positively) with greater non-verbal intelligence skills. In contrast, the Flanker task may be overall less cognitively challenging than DCCS, with an overall faster speed of responding. Here, individually smaller conflict scores might depend more on lower speed costs of the incongruent flanker condition compared to the congruent. In turn, greater costs would be associated (negatively) with non-verbal intelligence. Ultimately, it is very hard to disentangle these explanations on the basis of difference scores alone, because they lack a true baseline for overall difficulty or cognitive load. We have now included these suggestions in the discussion of unexpected / intriguing findings, and our evaluation of difference scores (pp. 20).

We now state:

In addition, some unexpected correlations also emerged, again with difference score; the DCCS conflict score was positively associated with RSPM, whereas the Flanker Conflict score correlated negatively with it. This may suggest a divergence between the two (reaction time based) difference scores and how they relate to individual differences in non-verbal intelligence. Although we can only speculate on interpretations, it is possible that the differences in relations to RSPM depend on different levels of overall cognitive load and average speed imposed by the two tasks. Ultimately, it is very hard to disentangle alternative explanations on the basis of difference scores alone, because they lack a true baseline for overall difficulty or cognitive load.

7. The summary of results on page 15 does not match the tables. Non-verbal IQ was associated with switch scores in Guatemala and the Philippines, but not in South Africa.

Response: This has been corrected.

Reviewer #2: Tests of executive functions (EF) are most frequently evaluated based on different scores describing reaction time and/or accuracy of performances. Results and interpretations depend on the chosen scoring method. The presented study evaluates two scoring methods of inhibition and cognitive flexibility assessed via the NIH Toolbox across three large-scale samples from different countries. Results highlight differences in these methods and possible implications are discussed.

The right choice of measurement is an important issue researchers face when conducting studies and evaluating data. Therefore, comparing strengths and weaknesses of possible scoring methods can prove valuable for future studies. However, I find the focus of the presented comparison to miss critical aspects enabling that choice and have a few additional concerns.

Major Points:

1. I find the introduction and discussion of the scores regarding their theoretical and specific cognitive implications to be a little scarce. Given that the study’s aim is to compare the suitability of scores across populations the theoretical conclusions enabled based on both kinds of scores should also be considered. This is touched upon in one paragraph of the discussion but could be further expanded and introduced. Considerations about conclusions regarding cognitive functionality, etc. provide valuable information to researchers since the study of interindividual differences might not always be the focus for applications.

Response: We thank the Reviewer for this call to expand on the theoretical and cognitive framework underpinning difference scores, as opposed to combined scores (such as the two-vector approach). In response to both reviewers, we have extended this in the introduction (p. 3-4) and in the discussion (p.20). In summary, what we say is as follows:, 

In the introduction: The current study compared an approach based on difference scores (as criticised by 6,10) and an approach combining accuracy and speed (12,13), using data from three diverse low and middle income birth cohorts. An exhaustive treatment of all approaches that treat accuracy and speed separately (e.g., difference scores, 8), as opposed to combining them (e.g., composites, latent factors, 12, 13 ), is beyond the scope of a single manuscript. However, a comparison could be really informative in the context of data that are unique or unusual in representing populations outside the Global North.

In the discussion: A key broader point emerging from the comparison of two scoring methods, is that, if both accuracy and reaction time convey unique information to EF ability, they could be combined. The benefit of combined scores applies to the broader class of combined scores like the two vector-scores solution by NIH Toolbox, but also other approaches, such as latent variable models (12,13). While a comparison of all combined methods was beyond the scope of a single manuscript, it is really worth noting that, as a class of methods, they outperform difference scores along many dimensions. 

We thank the reviewer for the second recommendation presented at this point. We are not entirely certain about the meaning of this statement and interpret it to mean that considerations regarding cognitive functionality should be reflected in our discussion, to provide information to researchers who do not often focus on interindividual differences. We agree with this point and we have now included this into our discussion. 

In the discussion: A further benefit is that individual differences across multiple underlying cognitive processes feeding into both accuracy and speed are well reflected by combined scores. This is important for researchers interested in capturing cognitive underpinnings of individual differences.

2. I do not fully understand the reasoning behind the correlation between scores and non-verbal IQ. EFs are linked to general intelligence. However, I would not use an associated construct, i.e., intelligence, to examine whether scores validly reflect a different construct, i.e., EF, as the authors seem to conclude.

Response: We acknowledge that the two constructs are related at a conceptual level. However, empirically, distinct EF scoring methods from data collected using the NIH toolbox on the one hand, and validated robust measures of non-verbal intelligence collected with a separate internally valid tool (RSPCM) on the other have not been previously correlated. Our analysis attempted to test the hypothesized theoretical relationship of EF and (non-verbal) intelligence, with the novel measures collected here.

3. The results of the Guatemala cohort, as the authors hypothesize themselves, are likely due to participants performing the task wrong. For one sixth of participants the switch cost score could not be computed. Solely based on the computation criteria of the scores the DCCS Computed Score can still be calculated. However, I am unsure whether this should be the case since results reflect performance in a different task, i.e., there is no switching component. Results in and interpretations about this cohort are thus substantially influenced by a group of participants who did not perform the task. Maybe as control, results without this subgroup should be considered.

Response: This is an interesting point. However, we do not feel that those participants who scored ‘7’ were due to participants completing the task wrong, but more that these participants found the task most challenging. All participants successfully passed the practice trials, as stated in the discussion. We do not think a sub-analysis removing these participants would add to the findings of the paper, as removing these participants would be removing valid scores (as they did demonstrate they understood the task during the practice trials), of participants who found the task most challenging and thus are an important part of the distribution of data for this task, and important to portray. 

Minor Points:

1. A suggestion related to my first major point: A key aspect of the study is the benefit provided by scores including both accuracy and speed information rather than just one of the above. A lot of the differences found between scores in this study can be boiled down to this distinction. Although the chosen scoring methods are examples of both cases, I feel that the discussion could benefit from a more general discussion of combined speed and accuracy measures.

Response: We agree with the Reviewer entirely and we thank them for the opportunity of clarifying this, both in the introduction (p. 3-4) and the discussion (p. 20-21), as well as linking to other scoring methods that, like the combined approach here were recommended by the other Reviewer, 

2. Although the sample it described elsewhere a little more information like mean/ median age and SD or gender distributions would be informative.

Response: The age range for participants is included in the cohort descriptions and we have also added a description of the age and gender distributions to the beginning of the results section. 

3. The descriptions of the task procedures are rather unspecific. For instance, in the Flanker task, where are the icons presented and are they mapped to different hands? In the DCCS, is stimulus presentation sequentially?

Response: We added more details to the description of each task and included pictures of the practice trails from each task.

4. Table 1 is missing information on which information is presented in parentheses (SD/ SEM/ IQR)?

Response: We clarified the information presented in Table 1. 

5. The bins of the histograms of Figures 2 and 3 for the difference scores differ between cohorts and seem coarse given that they present reaction time data which often depict and find much smaller differences on the level of milliseconds.

Response: We revised the figures to have consistent bins across cohorts. The values represent the difference in reaction time between trial types, which are generally on the scale of milliseconds, but the x-axis needs to be in seconds to capture outlier values, which are important to capture for the objective of the study.

6. The labels of the graphs describing the difference scores could reflect more clearly what is depicted, i.e., difference between respective conditions in seconds.

Response: We have changed the labeling of the y axis and added a footnote to the figures for clarity.

7. Could you explain in more detail how the reaction times are rescaled in the computed scores/ state more clearly which score from 0-5 represents which direction of reaction times? I was a little confused by the inversion appearing in the discussion (0 = slow, 5 = fast)

Response: The reaction times are log transformed to improve normality of the distribution and then truncated to a minimum of 500ms and maximum of 3000 ms, as needed. The log values are then rescaled so the reaction time vector ranges from 0-5 (like the accuracy vector). Because faster response times correspond to “better” performance, log scores are reversed during the rescaling so that smaller log scores are at the upper end of the 0-5 range, while larger log values are at the lower end. We clarified this in the NIH Computed Score section of the methods.

---

## [Decision Letter · Decision Letter 1]

1 Jun 2023

PONE-D-22-02319R1Contrasting speed and accuracy approaches to measure executive functions in three low-and middle-income countriesPLOS ONE

Dear Dr. Wray,

Thank you for submitting your manuscript to PLOS ONE. After careful consideration, we feel that it has merit but does not fully meet PLOS ONE’s publication criteria as it currently stands. Therefore, we invite you to submit a revised version of the manuscript that addresses the points raised during the review process.

You will see that although the reviewers were largely satisfied with the amendments to the manuscript, there are still some methodological points to be considered and addressed, before the manuscript is recommended for publication. I would, therefore encourage you to consider them carefully and address them accordingly.   

We look forward to receiving your revised manuscript.

Kind regards,

Theodoros M. Bampouras

Academic Editor

PLOS ONE

Journal Requirements:

Reviewers' comments:

Reviewer's Responses to Questions

**Comments to the Author**

1. If the authors have adequately addressed your comments raised in a previous round of review and you feel that this manuscript is now acceptable for publication, you may indicate that here to bypass the “Comments to the Author” section, enter your conflict of interest statement in the “Confidential to Editor” section, and submit your "Accept" recommendation.

Reviewer #1: (No Response)

Reviewer #2: All comments have been addressed

2. Is the manuscript technically sound, and do the data support the conclusions?

Reviewer #1: Partly

Reviewer #2: Yes

3. Has the statistical analysis been performed appropriately and rigorously? 

Reviewer #1: Yes

Reviewer #2: N/A

4. Have the authors made all data underlying the findings in their manuscript fully available?

Reviewer #1: No

Reviewer #2: Yes

5. Is the manuscript presented in an intelligible fashion and written in standard English?

Reviewer #1: Yes

Reviewer #2: Yes

6. Review Comments to the Author

Reviewer #1: The authors have been mostly responsive to my comments. One area where they were not is in response to my comment #6. The DCCS switch score and DCCS computed score are positively correlated in two cohorts (Guatemala, Philippines) but negatively correlated in South Africa. The authors replied that the negative association in South Africa was not statistically significant. However, it is marked with an asterisk in Table 5 (-.06*) indicating that the negative association was significant. The authors need to resolve this discrepancy and provide additional explanation for why the scores may have different associations across contexts.

I also disagreed with their response to Reviewer 2, point 3. They claim that participants who scored '7' were not completing the task wrong. However, sorting by color on all trials IS performing the task incorrectly and displays a misunderstanding of the task instructions. This is especially apparent since there was no variability in performance (i.e., all except 1 participant got all repeat trials incorrect). I agree that this necessitates a need for sensitivity analyses excluding these participants.

In the discussion, when describing the participants who scored 5 points on the Flanker, the authors state that this means the participants responded slowly but accurately. This raises a point worth mentioning that RT scores are only useful if all participants are following the same instructions (i.e., "respond as quickly as you can") -- which is hard to determine. This is problematic whether or not you are using RT difference scores or combined accuracy/RT scores and may be a bigger issues in cultures where accuracy is considered to be more important than speed.

Another limitation worth mentioning when comparing NIH toolbox computed scores to latent variable scores is that NIH Toolbox weighs each item that goes into the vector score equally, whereas a latent variable score would empirically determine how strongly to weight each item. This could lead to more precise quantification of individual differences.

Reviewer #2: (No Response)

7. PLOS authors have the option to publish the peer review history of their article (what does this mean?). If published, this will include your full peer review and any attached files.

Reviewer #1: No

Reviewer #2: No

---

## [Author Response · Author response to Decision Letter 1]

18 Jul 2023

1. The authors have been mostly responsive to my comments. One area where they were not is in response to my comment #6. The DCCS switch score and DCCS computed score are positively correlated in two cohorts (Guatemala, Philippines) but negatively correlated in South Africa. The authors replied that the negative association in South Africa was not statistically significant. However, it is marked with an asterisk in Table 5 (-.06*) indicating that the negative association was significant. The authors need to resolve this discrepancy and provide additional explanation for why the scores may have different associations across contexts.

Response: We thank the reviewer for their thorough review and pointing out this discrepancy. Our comment about the correlation not being statistically significant was misplaced (the subject of the following comment #7 was about the correlation between non-verbal IQ and DCCS switch cost in South Africa, which was nonsignificant; we believe we made a typographical error). 

The correlation between the DCCS switch score and DCCS computed score in South Africa is negative and statistically significant (r = -0.06, p = 0.02). We checked this analysis and confirmed that it is presented accurately in the table and text. 

 As for why the DCCS switch score and computed score are positively correlated in Guatemala and the Philippines and negatively correlated in South Africa, we can offer some suggestions that would require further testing and validation elsewhere (please see line 310 for a brief account for readers). A number of sample and data characteristics distinguish the South African cohort from the other two samples. First, from a demographic point of view, the South Africa participants are younger and reached higher educational levels. Second, from the point of view of data distributions, the South Africa participants obtained slightly higher DCCS absolute raw scores, high accuracy, large switch costs and high computed scores. High accuracy and large switch costs are important in understanding this discrepancy, because the computed score calculation takes into account RT only when a present accuracy criterion is met. If for South Africa individual participants higher accuracy was combined with relatively slow reaction time on switch trials, this would result in a computed scores that is primarily influenced by speed and is lower relative to participants with high accuracy and fast switch trials, which could in turn account for a negative correlation between switch score and computed score. The other two samples, in contrast, did not exhibit overall high accuracy, so that computed scores were more influenced by accuracy and less influenced solely by (less large) reaction time differences between trial types. 

2. I also disagreed with their response to Reviewer 2, point 3. They claim that participants who scored '7' were not completing the task wrong. However, sorting by color on all trials IS performing the task incorrectly and displays a misunderstanding of the task instructions. This is especially apparent since there was no variability in performance (i.e., all except 1 participant got all repeat trials incorrect). I agree that this necessitates a need for sensitivity analyses excluding these participants.

Response: As requested, we conducted a sensitivity analysis excluding observations who had zero correct repeat trials (n = 182 from Guatemala) and examined correlations with the NIH computed scores, difference scores, and non-verbal IQ. 

Observations with a score of zero on the repeat trials do not have a DCCS switch score. As is noted on page 11 line 251, “In addition, DCCS switch cost scores could not be calculated for 190 (15.5%) Guatemalan, 13 (1.0%) Filipino, and 12 (0.9%) South African participants.” These observations are not included in the calculation of the correlation between the DCCS switch score and DCCS computed score and therefore the correlation in the sensitivity analysis is the same as the original manuscript. Correlations between the other measures changed slightly once the observations were excluded but were not meaningfully different (see tables below for comparison). 

Table 3. Associations between each of the measures in Guatemala. 

Guatemala RSPM Flanker Conflict

Score DCCS Switch

Cost Flanker Computed Score

Flanker Conflict Score -.15** - 

DCCS Switch Cost Score .09** -.07** - 

Flanker Computed Score .45** -.37** .10** -

DCCS Computed Score .50** -.18** .21** .54**

Note: *p<.05 **p<0.01, RSPM: Raven’s Standard Progressive Matrices

Table 3. Sensitivity analysis. 

Guatemala RSPM Flanker Conflict

Score DCCS Switch

Cost Flanker Computed Score

Flanker Conflict Score -.17** - 

DCCS Switch Cost Score .09** -.07* - 

Flanker Computed Score .44** -.38** .10** -

DCCS Computed Score .49** -.14** .21** .55**

We suggest that this supports our interpretation that the excluded observations in Guatemala were low performers on the DCCS task and we would expect them to do poorly on other tasks. We expect that excluding observations with a zero repeat trial score would remove observations from the left tail of the score distribution on other tasks and because of this, correlations between tasks would be similar. Conversely, if zero repeat trial scores were not related to performance, we would expect that excluding observations for the sensitivity analysis would remove datapoints from the other score distributions at random, resulting in changes to the correlations between tasks, which was not observed. 

We do not think the manuscript would benefit from including this sensitivity analysis in the main text. However, we are happy to include it as supplementary material if the editor and reviewers think that it is useful for the reader.

3. In the discussion, when describing the participants who scored 5 points on the Flanker, the authors state that this means the participants responded slowly but accurately. This raises a point worth mentioning that RT scores are only useful if all participants are following the same instructions (i.e., "respond as quickly as you can") -- which is hard to determine. This is problematic whether or not you are using RT difference scores or combined accuracy/RT scores and may be a bigger issues in cultures where accuracy is considered to be more important than speed.

Response: We have added the following to the discussion.

“ This raises the question about the impact that task instructions have on participant response behaviours. For example, in cultures or age groups that prioritise accuracy over speed, the instruction “respond as quickly as you can” may be followed to a different extent. It is difficult to determine the extent to which participants follow task instructions and whether participants prioritise accuracy over speed or vice versa. However, this is a particularly important point to highlight given the diversity in culture and ages across our cohorts and may explain some of the differences observed across cohorts. This limitation would equally impact both reaction time scores and combined scores.”

4. Another limitation worth mentioning when comparing NIH toolbox computed scores to latent variable scores is that NIH Toolbox weighs each item that goes into the vector score equally, whereas a latent variable score would empirically determine how strongly to weight each item. This could lead to more precise quantification of individual differences.

Response: This is another valid limitation of the toolbox and we have added it to the discussion. 

“However, one advantage of a latent score over the NIH Toolbox approach is that latent scores empirically determine how strongly to weight each item, whereas each item is weighted equally in the NIH Toolbox score. As such the latent variable score may be a more precise measurement of individual differences.”

---

## [Decision Letter · Decision Letter 2]

7 Aug 2023

Contrasting speed and accuracy approaches to measure executive functions in three low-and middle-income countries

PONE-D-22-02319R2

Dear Dr. Wray,

We’re pleased to inform you that your manuscript has been judged scientifically suitable for publication and will be formally accepted for publication once it meets all outstanding technical requirements.

Kind regards,

Theodoros M. Bampouras

Academic Editor

PLOS ONE

Additional Editor Comments (optional):

Reviewers' comments:

Reviewer's Responses to Questions

**Comments to the Author**

1. If the authors have adequately addressed your comments raised in a previous round of review and you feel that this manuscript is now acceptable for publication, you may indicate that here to bypass the “Comments to the Author” section, enter your conflict of interest statement in the “Confidential to Editor” section, and submit your "Accept" recommendation.

Reviewer #1: All comments have been addressed

2. Is the manuscript technically sound, and do the data support the conclusions?

Reviewer #1: (No Response)

3. Has the statistical analysis been performed appropriately and rigorously? 

Reviewer #1: (No Response)

4. Have the authors made all data underlying the findings in their manuscript fully available?

Reviewer #1: (No Response)

5. Is the manuscript presented in an intelligible fashion and written in standard English?

Reviewer #1: (No Response)

6. Review Comments to the Author

Reviewer #1: (No Response)

7. PLOS authors have the option to publish the peer review history of their article (what does this mean?). If published, this will include your full peer review and any attached files.

Reviewer #1: No

---

## [Editor Report · Acceptance letter]

16 Aug 2023

PONE-D-22-02319R2 

Contrasting speed and accuracy approaches to measure executive functions in three low-and middle-income countries 

Dear Dr. Wray:

I'm pleased to inform you that your manuscript has been deemed suitable for publication in PLOS ONE. Congratulations! Your manuscript is now with our production department. 

Kind regards, 

on behalf of

Dr. Theodoros M. Bampouras 

Academic Editor

PLOS ONE